# Assessment of Three Automated Identification Methods for Ground Object Based on UAV Imagery

Ke Zhang [1], Sarvesh Maskey [2], Hiromu Okazawa [2,*], Kiichiro Hayashi [3,*], Tamano Hayashi [4], Ayako Sekiyama [2], Sawahiko Shimada [2] and Lameck Fiwa [5]

1. Faculty of Agriculture, Graduate School of Tokyo University of Agriculture, Tokyo 156-8502, Japan
2. Faculty of Regional Environment Science, Tokyo University of Agriculture, Tokyo 156-8502, Japan
3. Institute of Material and Systems for Sustainable, Nagoya University, Nagoya 464-8601, Japan
4. Faculty of Advanced Science and Technology of Regional Environment Science, Ryukoku University, Kyoto 612-8577, Japan
5. Faculty of Agriculture, Lilongwe University of Agriculture and Natural Resources, Lilongwe P.O. Box 219, Malawi
* Correspondence: h1okazaw@nodai.ac.jp (H.O.); maruhaya98–@imass.nagoya-u.ac.jp (K.H.); Tel.: +81-13-5477-2685 (H.O.)

**Abstract:** Identification and monitoring of diverse resources or wastes on the ground is important for integrated resource management. The unmanned aerial vehicle (UAV), with its high resolution and facility, is the optimal tool for monitoring ground objects accurately and efficiently. However, previous studies have focused on applying classification methodology on land use and agronomy, and few studies have compared different classification methods using UAV imagery. It is necessary to fully utilize the high resolution of UAV by applying the classification methodology to ground object identification. This study compared three classification methods: A. NDVI threshold, B. RGB image-based machine learning, and C. object-based image analysis (OBIA). Method A was the least time-consuming and could identify vegetation and soil with high accuracy (user's accuracy > 0.80), but had poor performance at classifying dead vegetation, plastic, and metal (user's accuracy < 0.50). Both Methods B and C were time- and labor-consuming, but had very high accuracy in separating vegetation, soil, plastic, and metal (user's accuracy ≥ 0.70 for all classes). Method B showed a good performance in identifying objects with bright colors, whereas Method C showed a high ability in separating objects with similar visual appearances. Scientifically, this study has verified the possibility of using the existing classification methods on identifying small ground objects with a size of less than 1 m, and has discussed the reasons for the different accuracy of the three methods. Practically, these results help users from different fields to choose an appropriate method that suits their target, so that different wastes or multiple resources can be monitored at the same time by combining different methods, which contributes to an improved integrated resource management system.

**Keywords:** UAV; NDVI; orthomosaic; classification; OBIA; machine learning; threshold

## 1. Introduction

Despite recent advances and development in Earth-observing satellites, temporal resolution and cloud cover are some of the obstacles present for many quantitative remote sensing applications, such as monitoring and detecting the dynamics of environmental systems. Since the 2010s, unmanned aerial vehicles (UAV) have been popular for various purposes, such as disaster relief, civil engineering surveys, pesticide spraying, and infrastructure inspections [1]. Compared to earth observation satellites such as Landsat, Terr, and SPOT, UAVs have advantages such as high mobility, high resolution, and low-altitude flight (unaffected by clouds), which enable them to achieve highly accurate and precise observation of ground objects.

Ground object identification using UAV imagery is helpful for an improved environmental and resource management system. For example, the occurrence of waste items all over villages, farmland, and natural parks have resulted in garbage management becoming a serious local environmental issue. Additionally, plastic pollution due to agricultural activities is an important source of pollution as plastics are difficult to quantify [2]. Furthermore, uncontrolled open dumping and burning pollutes water and soil, affects plants, increases vectors of disease, emits odors and greenhouse gasses into the atmosphere, and poses serious health risks to people working at open dumping sites [3,4]. Micro-plastics, formed when waste plastics are fragmented by photochemical, mechanical, and biological processes, contaminate aquatic ecosystems through passive or active ingestion by a wide range of organisms [5]. Environmental degradation, because of poor waste management, decreases the quality and quantity of forest, fisheries, and tourism resources. Such degradation has negative impacts on local industries, which in turn indirectly affects people's well-being [6–9]. To sustainably mitigate and monitor drivers of environmental degradation, including ground objects such as agricultural wastes, vegetation, soil, weak vegetation, plastic sheets and metals, requires transdisciplinary collaboration in identification and monitoring amongst societal stakeholders and researchers.

Vegetation canopy cover monitoring is another topic that can be benefited from precise ground object identification, providing important information for forestry management and ecosystem service surveys. Canopy cover (CC) is an easily measured characteristic that is an indicator of crop growth and an important parameter in crop simulation models, such as the Aqua Crop model [10]. Accurate and efficient CC estimation would allow improved scheduling and allocation of irrigation water [11]. Furthermore, identifying dead or weakened plants can help farmers to make better field management decisions. Therefore, identifying crop cover and weakened vegetation precisely and efficiently using UAV imagery is thought to be helpful to rural environmental management, agriculture development, and integrated resource management.

Ground object classification has been studied worldwide and can be achieved through different approaches. Vegetation indices have been extensively used to trace and monitor vegetation conditions such as health, growth levels, and water or nutrient stress [12]. Previous studies have shown that various spectral calculations based on visible and near-infrared reflectance data can reflect the growth status of vegetation [13]. Not only can the health condition of plants be monitored using NDVI, but discovery of the weakened vegetation, soil, and plastic and metal items is also possible, which have significantly different reflectance rates in the band ranges of red and near-infrared light [14]. Furthermore, PVC and metal items have lower NDVIs than that of soil because of a relatively higher reflectance in the red range and a lower reflectance of near-infrared range than soil [15]. Therefore, the NDVI threshold method has been used as one of the standards to classify the land cover on the fields of national parks, rural sociology, urban environmental engineering and ecology [15–19]. This research shows that the NDVI threshold method is a practical classifier for land use classification.

Other methods for land cover classification include machine learning approaches, which classify the image depending on appropriate training samples. The machine learning algorithm allows image diagnosis to be conducted in an automatic and efficient manner. One of the most common methods of machine learning classification based on red, green and blue (RGB) images, such as the orthomosaic constructed from aerial images, is to classify the pixels depending on their RGB values according to the training samples. Hassan et al. (2011) generated land use/land cover maps with UAV-obtained RGB images using the supervised classification algorithm (maximum likelihood) and achieved a 90% overall classification accuracy [20]. Hamylton et al. (2020) compared the classification results with UAV GCB images using the pixel classification, visual interpretation, and machine learning approaches, and the machine learning method showed the highest overall accuracy of 85% [21]. Shin et al. (2019) conducted classification of forest burn severity with UAV-obtained multispectral imagery using the maximum likelihood and

threshold methods and achieved overall accuracies of 89% and 71%, respectively [22]. These results showed that the pixel-based machine learning method could achieve very high accuracy in land cover classification.

Differing from traditional pixel-based classification methods, the object-based image analysis (OBIA) method first separates the image into segments which are small polygons constructed of several neighboring and similar-valued pixels [23]. Then, with appropriate training samples, the classification is performed by dividing the segments into different classes according to their shape, size, and spectral content [24]. Compared to traditional pixel-based classification methods, OBIA is thought to be accurate for hydrologic modeling and vegetation detection owing to its ability to detect the health status as well as the factors influencing the biological habitats in a rapid, accurate, and cost-effective manner [25]. The OBIA is one of the most popular classifiers for land cover classification and has been applied and verified worldwide in the fields of forestry, agriculture and oceanology [26–30]. However, no study has yet compared the classification accuracy of methods using the NDVI threshold, RGB image-based machine learning, and OBIA in rural areas

The automated identification methods have been applied in agriculture for batter field monitoring and management. Lanthier et al. (2008) conducted a comparative study between supervised pixel-oriented and OBIA classifications in a precision agriculture context using hyperspectral images to identify three different crop species (corn, peas and beans), and found out that the OBIA method achieved better performance, with a Kappa of 0.8268 [31]. Lebourgeois et al. (2017) analyzed and optimized the performance of a combined Random Forest classifier/OBIA approach and applied it to multisource satellite data to produce land use maps of a smallholder agricultural zone at five different nomenclature levels of the crops, achieving an overall accuracy of 91.7% and 64.4% for the cropland and crop subclass levels, respectively [32]. Zheng et al. (2019) presented the crop vision dataset for deep-learning-based classification and detection method for over 30 categories of crops and achieved an overall accuracy of over 99% [33]. However, these studies were only focused on vegetation monitoring and classification, instead of the overall environment including the non-vegetation objects, which also have an influence on better field management.

Unlike satellite remote sensing methods, UAV surveys can perform identification accurately with high resolution, and are also suited for small-scale research applications. These methods could provide societal stakeholders and researchers with a transdisciplinary approach to ground object identification, monitoring and decision-making abilities, contributing to sustainable community development. However, for this approach to be used by stakeholders involved in integrated resource management, the following should be made clear (1) What are the characteristics of the three methods in terms of the accuracy of identification of ground objects? (2) What are the advantages and disadvantages of the three methods for different ground objects? (3) what are the recommendations for the choice of different methods toward integrated resource management? Therefore, this study aimed at performing ground matter identification with three different methods (NDVI threshold, RGB image-based machine learning, and Object-based image analysis (OBIA) method), comparing the total overall accuracy, and discussing the characteristics and optimal classification for each method.

## 2. Materials and Methods

### 2.1. Study Site

The aerial surveys were conducted within the experimental field (total area: 3.2 ha) of Obihiro University of Agriculture and Veterinary Medicine located at 42.8688° N, 143.1725° E, at an altitude of 75 m. The area used for classification verification is shown surrounded by the red dotted line in Figure 1. The ground objects at the study site were vegetation, dead/weakened vegetation, soil, plastic multi-sheet, plastic blue-sheet, and metal pipes (Figure 2). The vegetation included crops such as wheat, pasture grass, pumpkin, and peanut. The dead/weakened vegetation in this study site was barley, which was near the harvest stage. If the methods discussed in this study could identify the weakened vegetation accurately, it would provide a

useful tool for precision agriculture by helping farmers detect growth problems among the crops. A multi-sheet is a thin, smooth, and translucent film used in agriculture fields to maintain the temperature and moisture of soil and is made of polyethylene, which is the same material as that from which plastic bags are manufactured, is barely biodegradable, and can cause problems not only for the natural environment, but also for the health of livestock and humans. The blue-sheet is made of the same material as the multi-sheet, but has a thick, rough, and blue-colored surface, which results in a difference in the spectrophotometry reflection characteristics of these two kinds of polyethylene products. Multi-sheet, blue-sheet, and metal pipes are materials that are often found in most open waste dumping sites, which means they have similar visual appearance and spectrophotometry reflection characteristics.

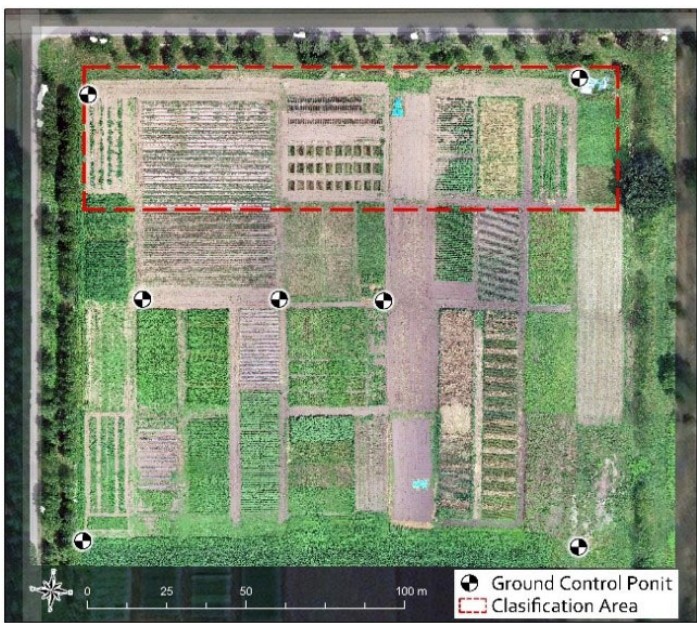

**Figure 1.** Aerial image of the experimental field with the classification verification site marked by the red dotted line.

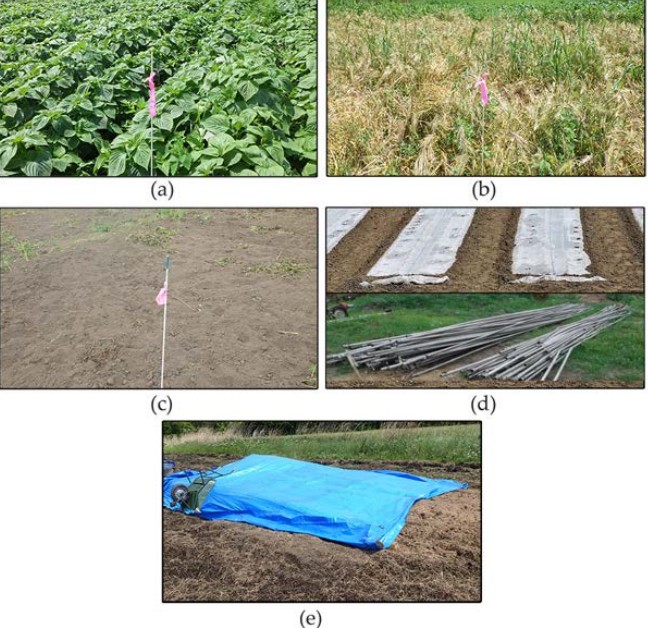

**Figure 2.** The ground objects at the study site: (**a**) vegetation; (**b**) dead/weakened vegetation; (**c**) soil; (**d**) multi-sheet and metal; (**e**) plastic blue sheet.

### 2.2. UAV Settings and Data Collection

Aerial surveys were conducted twice on August 1st, 2019. The lightweight UAVs and camera parameters used for both surveys are shown in Figure 3 and Table 1. The Phantom 4 Pro (DJI) was used to obtain the RGB images of the study site, and the Inspire 2 (DJI) equipped with a multispectral sensor and a sunlight sensor Sequoia (Parrot) was used to obtain the multispectral images (green band, 510–590 nm; red band, 620–700 nm; red-edge band, 715–775 nm; near-infrared band, 750–830 nm). A 10,000 mA mobile battery (Anker Power Core) was also attached to the Inspire 1 to power the multispectral sensor. Considering that the agriculture field had relatively simple ground objects and lacked the characteristic points that help to match images, the flight route used for both aerial surveys was a double grid to ensure successful image processing.

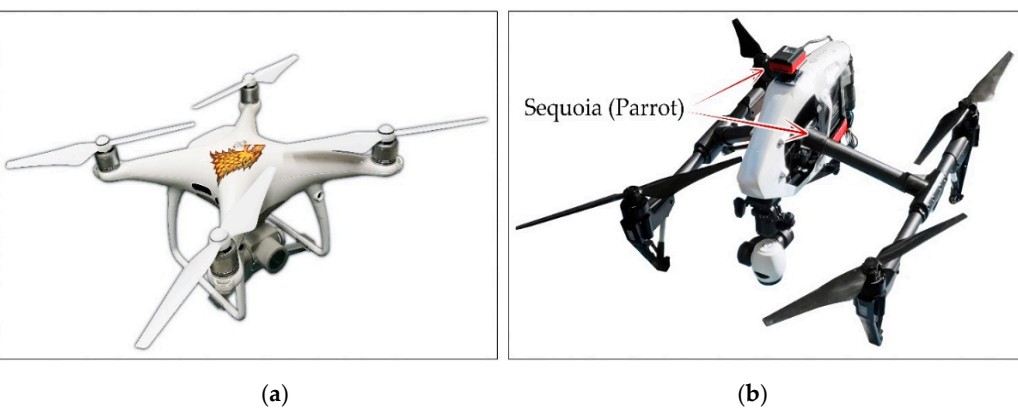

(**a**) (**b**)

**Figure 3.** Unmanned aerial vehicles (UAVs) used for the aerial surveys: (**a**) Phantom 4 Pro (DJI); (**b**) Inspire 1 (DJI) with Sequoia (Parrot).

**Table 1.** Unmanned aerial vehicle (UAV) and camera specifics.

|  | **RGB Imagery** | **Multispectral Imagery** |
|---|---|---|
| UAV model | Phantom 4 Pro (DJI) | Inspire 1 (DJI) |
| Total weight | 1375 g | 3400 g |
| Diagonal size | 350 mm | 581 mm |
| Maximum flight time | Approximately 30 min | Approximately 18 min |
| Camera type | 1 inch CMOS | Multispectral Sensor |
| Image size | 3840 × 2160 pixels | 1280 × 960 pixels |
| Angle of view | 84° | 74° |
| Top overlap rate | 80% | 80% |
| Side overlap rate | 80% | 80% |
| Camera angle | 75° from horizon | 90° degrees from horizon |
| Flight height | 50 m | 40 m |
| Ground resolution | 1.6 cm/pixel | 6.2 cm/pixel |

As shown in Figure 1, seven ground control points (GCPs) were selected within the experimental field. The position information of the seven GCPs was obtained by a Global Navigation Satellite System (GNSS) device Hiper V (TOPCON).

### 2.3. Structure from Motion Workflow

The Structure from Motion (SfM) technology can reconstruct the 3D structure of the object surface based on multiple, overlapping images taken by a moving camera. In the present study, the SfM process of the UAV images was conducted using Agisoft Metashape Professional Edition (ver. 1.8.0, Agisoft). The image processing workflow is illustrated in Figure 4. After obtaining the aerial images using UAVs and importing them into the software, a tie point cloud was generated by aligning the images and finding the characteristic points existing in the overlapping areas between the images. Then, the

position information of the GCPs obtained from the GNSS measurements was imported into the software and matched with the anti-aircraft signals in the images, which corrected the tie point cloud to the accurate geographic location. Based on the initial process results, a dense point cloud was generated based on depth maps calculated using dense stereo matching. Because generating the dense cloud was the most time-consuming step of the SfM process and the higher the density of the point cloud, the more complicated the subsequent calculations would be, in the present study, a medium-quality dense cloud was generated. Based on the dense cloud information, a 3D polygonal mesh was constructed by connecting the points with polygonal surfaces. After the surface model was constructed, a texture model was created by extracting the RGB color value and calibrating the brightness and white balance, giving the 3D model the same visual appearance as that of the actual object. Finally, based on the texture model, the RGB and multispectral (red and near-infrared) orthomosaics were exported. The ground resolution of the final products of the SfM procedures was 1.6 cm for panchromatic and 4.2 cm for multispectral products. The NDVI raster was then calculated using the Raster Calculator geo-processing tool in ArcGIS Pro (ver. 2.4.1, Esri).

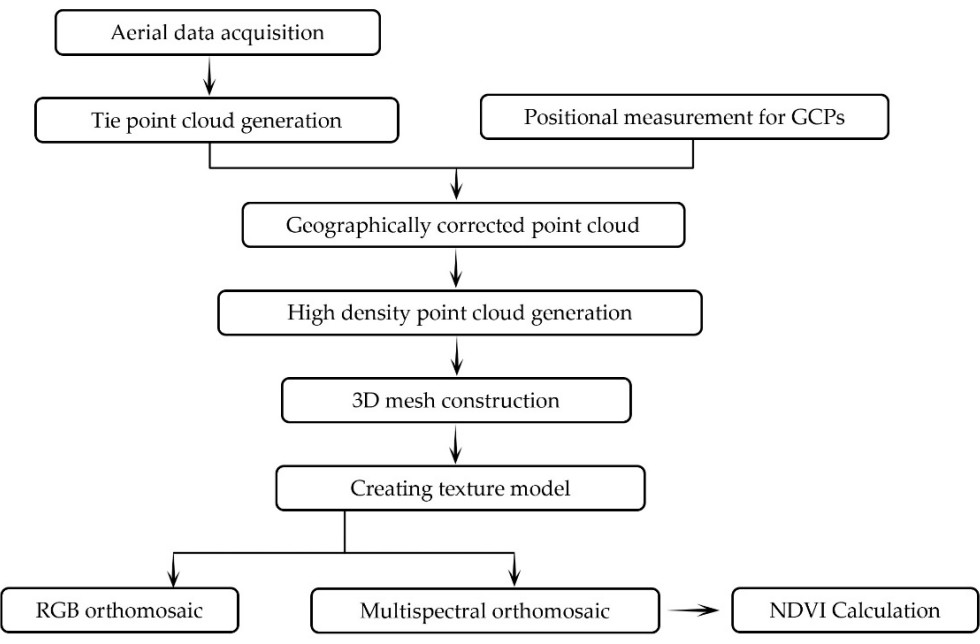

**Figure 4.** Workflow for the Structure from Motion (SfM) process.

*2.4. Classification Procedures*

As mentioned in the introduction, the NDVI is an index mostly used to evaluate the health condition of plants. The higher the NDVI value, the healthier the vegetation. Some objects other than plants also have specific NDVI values. The reflectance characteristics and the ranges of the NDVI values of these objects have been discussed by many previous studies [34–39]. Although the NDVI values and spectral characteristic of the same object are varied because of the differences in the used sensor or the ground resolution of the remote sensing data, there is an agreement that the dense and healthy vegetation has the NDVI value of more than 0.2 due to the extremely high reflectance of the near infrared band of light and the relatively low reflectance of the red light [40], and the NDVI value of the bare dry soil is around 0 due to the similar reflectance of the red light and the near infrared band of light [41]. Although the weakened vegetation has fewer chloroplasts, which leads to a lower reflectance of near infrared wave of light, the leaves are still conducting photosynthetic reaction. Therefore, the NDVI of the weakened or dead vegetation used in this study is lower than the normal vegetation, but slightly higher than the bare soil. The plastic materials have different reflectance features due to the coating color or transmittance

of light. However, the mean reflectance of red light (0.0375) from multiple plastic material is higher than the near infrared band of light (0.299), leading to a negative value of NDVI [35]. Same spectral features have been found on commonly used metals, such as Aluminum, iron and their alloys [42,43]. Based on this characteristic of NDVI, the NDVI thresholds were used to classify the study area. The optimal significant figures of the NDVI threshold for the classification of ground objects have been shown to be to the first decimal place, which was used in the NDVI threshold method in the present study (Table 2). The classification using this method was conducted within ArcGIS Pro (ver. 2.4.1, Esri). First, the NDVI raster was imported into the software. Then, the classes between the NDVI thresholds were extracted as independent raster layers using the Extract by Attributes geo-processing tool. To assign the attribute value for each class raster, the layers were processed using the Int tool, after which the vegetation layer was assigned the class number one, the soil layer as two, the dead/weakened vegetation layer as three, the multi-sheet and metal layer as four, and the blue-sheet layer as five. Additionally, the NDVI threshold of the blue-sheet class could not be defined since the NDVI value of the blue-sheet ranges from −0.1 to 0.1, which was included in both the soil class and the weakened vegetation class. Finally, the five layers were processed using the Mosaic to New Raster geo-processing tool, and a raster including the five classes of the entire study area was generated.

**Table 2.** Normalized difference vegetation index (NDVI) threshold values for the different classes.

| Class | Multi-Sheet and Metal | Soil | Weakened Vegetation | Vegetation |
|---|---|---|---|---|
| **NDVI threshold** | −0.3 to −0.2 | −0.2 to 0.0 | 0.0 to 0.2 | 0.2 to 1.0 |

The RGB image-based machine learning method uses the interactive Supervised Classification function of ArcGIS Pro. First, the RGB orthomosaic was input to the software, and a pyramid of the orthomosaic was built to achieve the optimal interactive performance. Next, five empty shape-file (polygon) layers for the classes (vegetation, soil, dead/weakened vegetation, multi-sheet and metal, and blue-sheet) were created and approximately 10–20 training samples for each class were manually distributed within the polygons. The number of training samples varied because the areas occupied by the different classes in the study area were not equal. To achieve the optimal classification result, all the training samples were determined at the pixel level, which means the error range was less than 2 cm. Therefore, although the required input data and operation steps for this method were simpler than those required for the NDVI threshold method, manual determination of the training samples was quite time-consuming. Finally, the maximum likelihood classification was performed on the orthomosaic layer of the study area based on the RGB values of the training samples.

The OBIA can classify image objects by dividing the entire image into small segments according to their shape, size, and spectral content. The software used for this method was eCognition Developer (ver. 9.0, Trimble). First, a new project including the RGB orthomosaic, red band reflectance orthomosaic, near-infrared reflectance orthomosaic, and NDVI raster was created in the software, and was displayed as one RGB-mixed layer in the workspace, where the near-infrared was displayed as green and NDVI as red. Next, the mixed image was separated into multiple segments using the Multiresolution Segmentation tool. The scale factor, which decides the average size of the segments and is commonly set between 100 and 150 for high resolution UAV images [25], was set to 100 in the present study to obtain the classification result as precisely as possible. The result of segmentation is shown in Figure 5. Then, five classes were created inside the Class Hierarchy window and 80 to 200 segments for each class were selected as training samples using the Select Samples tool. Finally, the classification was conducted according to the mean RGB value, mean brightness, standard deviation RGB, position, and shape of the mixed layer.

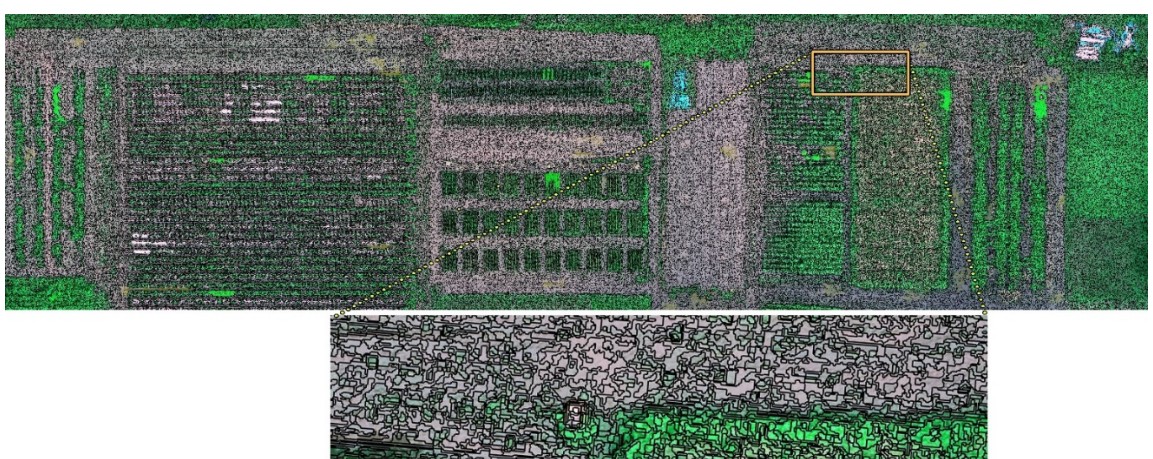

**Figure 5.** Segmentation result (eCognition Developer 9.0, Trimble).

*2.5. Accuracy Assessment*

The accuracy assessments for the classification results of the three methods were performed with ArcGIS Pro. First, the orthomosaic and classification raster from the classification methods were input into the software. Next, a point shapefile with 1000 assessment points was created using the Creating Accuracy Assessment Point geo-processing tool (Figure 6). The attribute table of the created point shapefile included both the Ground Truth field, which is the reference value, and the Classification Field, which is the test value. The reference value was determined by visual judgement, by zooming into the point position and deciding manually to what class the point belonged. The visual judgement was also conducted at the pixel level, which means the tolerance of error was less than 2 cm. The test value was extracted from the classification raster using the Extract Values to Points geo-processing tool. Finally, the accuracy assessment for each classification result was conducted using the Compute Confuse Matrix geo-processing tool, by calculating the user's accuracy, the producer's accuracy, and the Kappa coefficient.

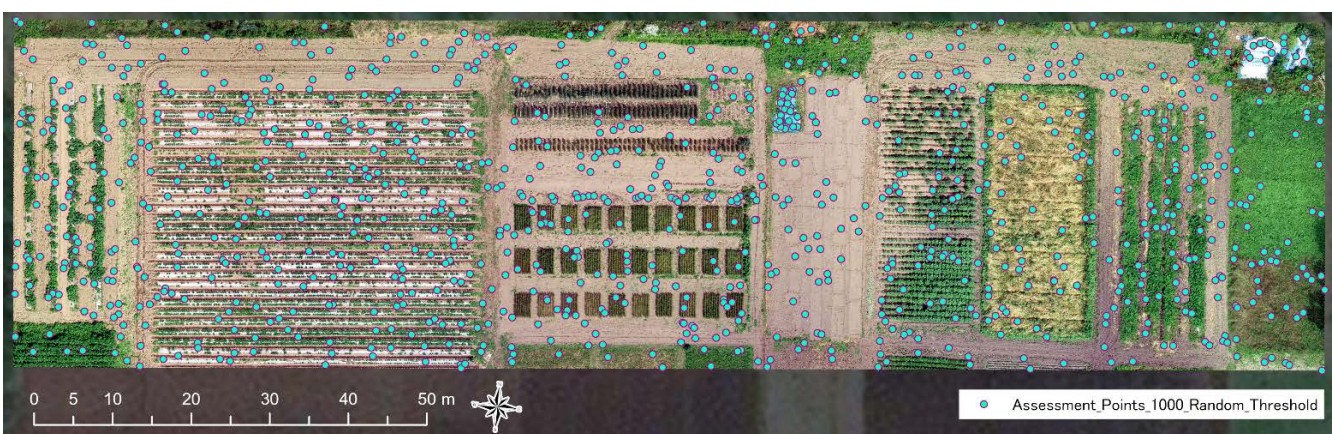

**Figure 6.** One thousand assessment points used for the accuracy assessment of the classification results.

**3. Results**

*3.1. UAV Mapping Products*

Figure 7 shows the RGB orthomosaic, red band orthomosaic, near-infrared orthomosaic, and NDVI raster generated from the UAV image, and the mixed RGB image in

eCognition Developer, which were the photogrammetry products used for the subsequent classification. The characteristic of each product was the basis for the different performances of the three methods.

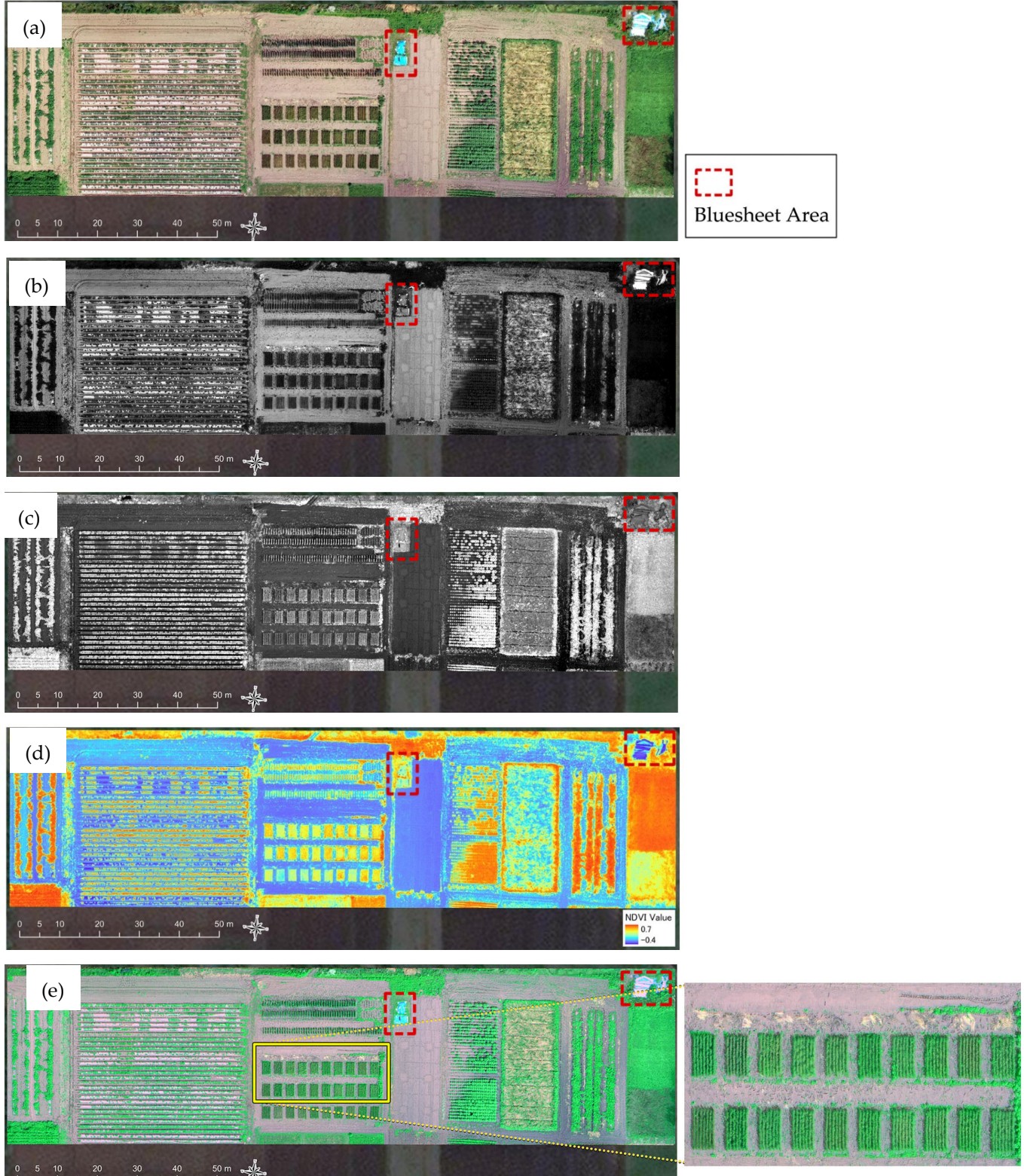

**Figure 7.** Mapping products generated from the unmanned aerial vehicle (UAV) images: (**a**) red-green-blue (RGB) orthomosaic; (**b**) red band orthomosaic; (**c**) near-infrared orthomosaic; (**d**) normalized difference

vegetation index (NDVI) raster; (**e**) mixed image in eCognition Developer.

As shown in Figure 7a, in the RGB orthomosaic, the vegetation, multi-sheet and metal, and blue sheet had characteristic RGB values that could be expressed clearly and distinctly. However, the RGB values of soil and weakened vegetation were close to each other, resulting in a similar visual appearance. Specifically, the average RGB values of soil were 223, 209, and 190, respectively, whereas those of the weakened vegetation were 217, 204, and 174, respectively, for ten randomly selected sample pixels. This fact suggested that the RGB orthomosaic had a disadvantage in distinguishing soil and weakened vegetation.

In contrast, as shown in Figure 7b, the red band orthomosaic could distinguish soil and weakened vegetation, but distinguishing between the weakened vegetation and the multi-sheet and metal was difficult because they all had a high reflectance rate of red light, as was distinguishing between the blue-sheet and soil because they both had an intermediate reflectance rate of red light.

As shown in Figure 7c, the vegetation area was clearly visible at the near-infrared orthomosaic, because the chlorophyll in healthy vegetation strongly reflects the near-infrared wavelength and appears as fluorescence in the near-infrared image. Even for the weakened vegetation, a small amount of chlorophyll still produced a visible fluorescence. This fact makes the near-infrared band an important indicator for vegetation in the remote sensing field. However, objects other than vegetation have no significant reflectance characteristic at the near-infrared, which resulted in the similar appearance of soil, multi-sheet, metal, and blue-sheet in the near-infrared orthomosaic. Because NDVI is a normalized value of the difference between the reflectance of the red and near-infrared bands, the NDVI raster shows more features of different objects that have specific characteristics in terms of red or near-infrared band reflectance.

As shown in Figure 7d, the NDVI raster clearly distinguished between soil, vegetation, and multi-sheet and metal. However, the weakened vegetation and blue-sheet had similar intermediate NDVI values as that of soil because the former had a high reflectance at both the red and near-infrared bands, and the latter had a low reflectance at both bands. Based on these findings, the RGB and multispectral orthomosaics had both advantages and disadvantages for identifying different ground objects. Therefore, a mixed layer was prepared in eCognition Developer to maximize the strength of each kind of data. As shown in Figure 7e, different from the orthomosaic, the mixed layer clearly displayed the weakened vegetation, and different from the multispectral orthomosaics and NDVI, it also clearly displayed objects with specific RGB values, such as the blue sheet.

### 3.2. Comparison of Classification Results

The classification results of the three methods are shown in Figure 8, and the Pixel percentage of the classes for each classification method is shown in Figure 9. Due to these results, the NDVI Threshold Method failed to identify the blue sheets, and tended to classify the vegetation and soil as the weakened vegetation. Contrastingly, the RGB Machine Learning Method classification result had the highest pixel percentage for the vegetation and the lowest pixel percentage for the weakened vegetation, showing its strength in identifying healthy vegetation and a weakness in identifying the weakened vegetation. On the other hand, as shown by the red square in Figure 8c, the OBIA method was able to identify the weakened vegetation which the RGB Machine Learning Method failed to recognize.

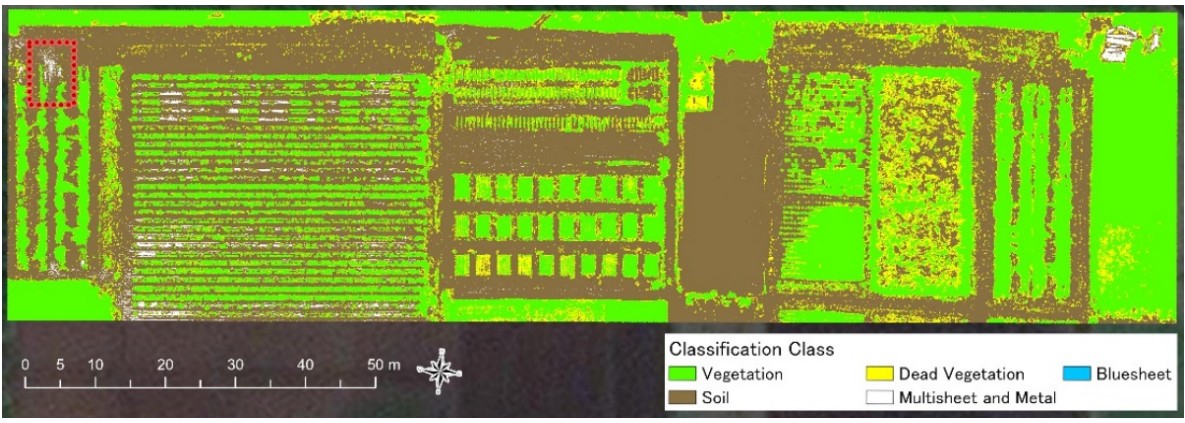

(**a**)

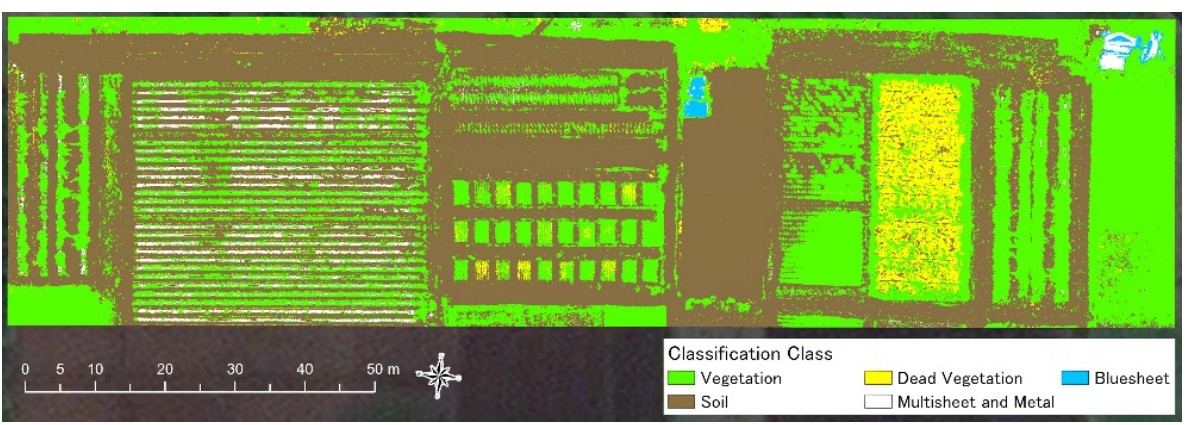

(**b**)

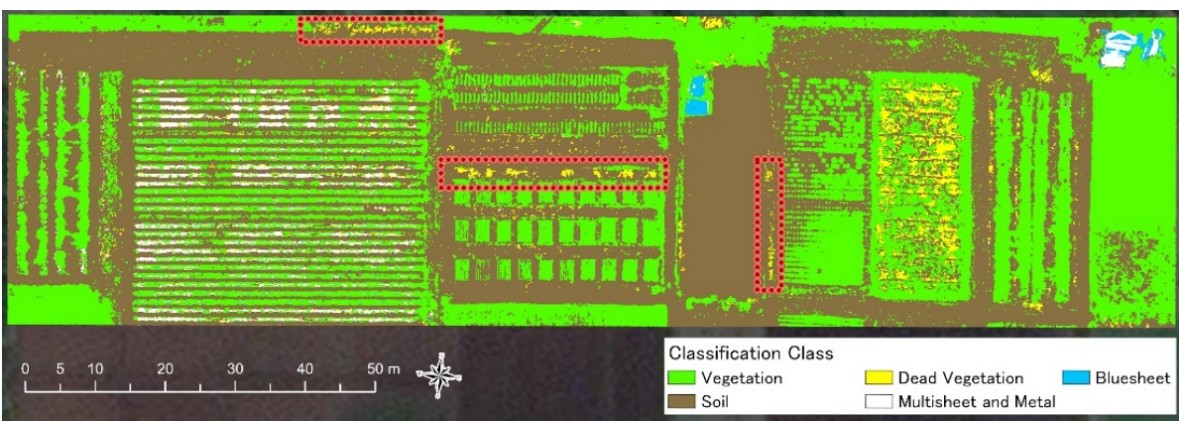

(**c**)

**Figure 8.** Classification results: (**a**) normalized difference vegetation index (NDVI) threshold method; (**b**) red-green-blue (RGB) image-based machine learning method; (**c**) object-based image analysis (OBIA) method.

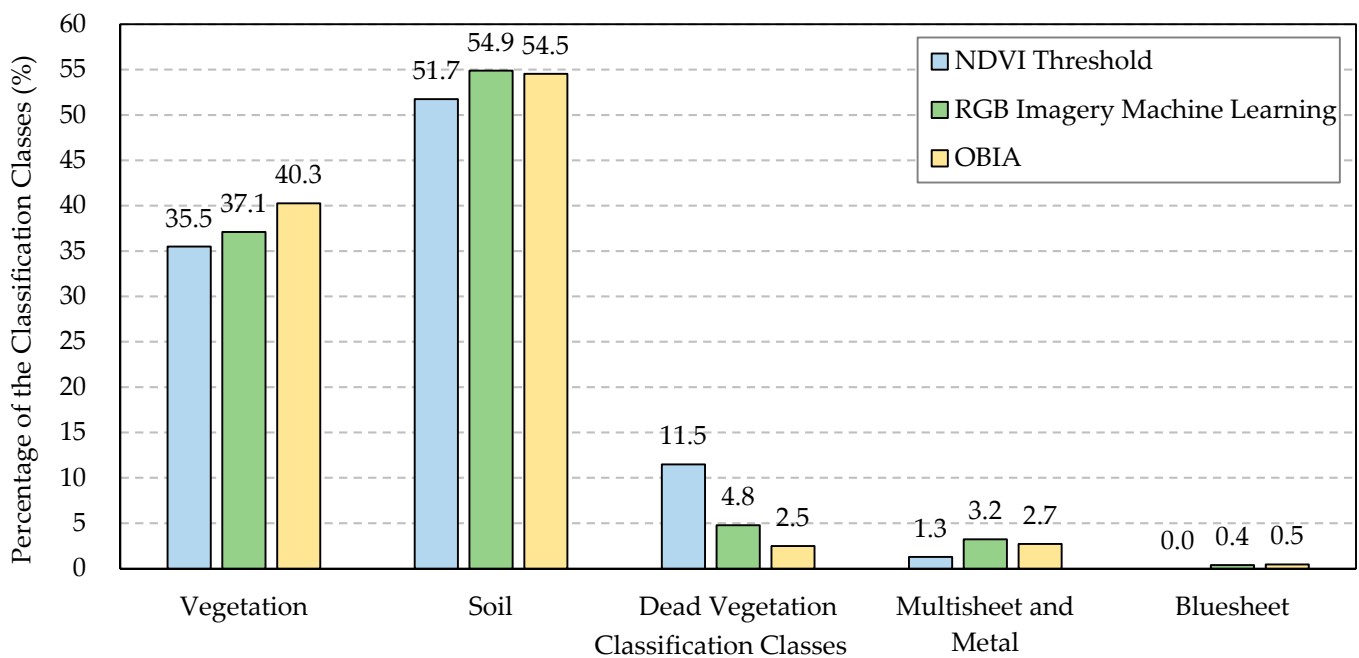

**Figure 9.** Pixel percentage of the classes for each classification method.

*3.3. Accuracy Evaluation of the Classification Methods*

The accuracies of the three methods were evaluated using the confusion matrix calculated with ArcGIS Pro. The confusion matrix (also known as error matrix) is a commonly used evaluation tool for classification verification. In the confusion matrix tables used in the present study, each row represents the classified results, whereas each column represents the reference results (the true value). For example, the first row of "Vegetation" in Table 3 lists the values 279, 33, 6, 5, 5, and 328, which means that among all the 328 points that were classified as "Vegetation" by this method, 279 points were within the vegetation area of the study site, whereas 33 points were in the soil area, indicating that they were misclassified. Similarly, the first column of "Vegetation" in Table 3 lists the values 279, 6, 32, 0, 0, 317, meaning that among the 317 points that should have been classified as vegetation, only 279 points were correctly classified by this method, whereas six points were misclassified as soil. Three indicators were calculated based on the confusion matrix and are presented in Table 3a–c: the user's accuracy, the producer's accuracy, and the overall Kappa index. The user's accuracy shows a false positive, meaning that the classification result was positive, whereas it should have been negative. For example, the user's accuracy for "Vegetation" in Table 3 was 0.851, meaning that among all the points that had been classified as vegetation by this method, only 85.1% were correct. Similarly, the producer's accuracy shows the false negative, meaning that the classification result was negative, but the correct answer should have been positive. For example, the producer's accuracy for "Vegetation" in Table 3 was 0.880, which means that among all the points that should have been classified as vegetation, only 88.0% were classified correctly by this method. The Kappa index, also known as Cohen's Kappa coefficient, is a statistical indicator used for conformance testing. The value of the Kappa index ranges between −1 and 1. The higher the Kappa index, the higher the classification accuracy.

Table 3a presents the confusion matrix of the NDVI threshold method. The overall Kappa index of this method was 0.576, which is considered "fair to good" [31]. The confusion matrix of the RGB image-based machine learning method is presented in Table 3b. The overall Kappa index of this method was 0.798, which is considered as "excellent". Table 3c presents the confusion matrix of the OBIA method. The overall Kappa index of this method was 0.793, which was close to that of the RGB image-based machine learning method and was also considered "excellent".

Table 3. Confusion matrix for the three classification methods.

| **(a) Confusion matrix for the normalized difference vegetation index (NDVI) threshold method** | | | | | | | | |
|---|---|---|---|---|---|---|---|---|
| **Class Name** | **Vegetation** | **Soil** | **Weakened Vegetation** | **Multi-Sheet** | **Blue-Sheet** | **Total** | **User_ Accuracy** | **Kappa** |
| Vegetation | 279 | 33 | 6 | 5 | 5 | 328 | 0.851 | |
| Soil | 6 | 426 | 58 | 18 | 6 | 514 | 0.829 | |
| Weakened vegetation | 32 | 59 | 29 | 2 | 23 | 145 | 0.200 | |
| Multisheet and Metal | 0 | 6 | 1 | 6 | 0 | 13 | 0.462 | |
| Bluesheet | 0 | 0 | 0 | 0 | 0 | 0 | 0.000 | |
| Total | 317 | 524 | 94 | 31 | 34 | 1000 | 0.000 | |
| Producer_accuracy | 0.880 | 0.813 | 0.309 | 0.194 | 0.000 | 0.000 | 0.740 | |
| Kappa | | | | | | | | 0.576 |
| **(b) Confusion matrix for the red-green-blue (RGB) mage-based machine learning method** | | | | | | | | |
| **Class Name** | **Vegetation** | **Soil** | **Weakened Vegetation** | **Multi-Sheet** | **Blue-Sheet** | **Total** | **User_ Accuracy** | **Kappa** |
| Vegetation | 300 | 27 | 5 | 0 | 0 | 332 | 0.904 | |
| Soil | 11 | 486 | 50 | 7 | 0 | 554 | 0.877 | |
| Weakened vegetation | 4 | 10 | 36 | 0 | 0 | 50 | 0.720 | |
| Multisheet and Metal | 1 | 1 | 3 | 24 | 1 | 30 | 0.800 | |
| Bluesheet | 1 | 0 | 0 | 0 | 33 | 34 | 0.971 | |
| Total | 317 | 524 | 94 | 31 | 34 | 1000 | 0.000 | |
| Producer_accuracy | 0.946 | 0.927 | 0.383 | 0.774 | 0.971 | 0.000 | 0.879 | |
| Kappa | | | | | | | | 0.798 |
| **(c) Confusion matrix for the object-based image analysis (OBIA) method** | | | | | | | | |
| **Class Name** | **Vegetation** | **Soil** | **Weakened Vegetation** | **Multi-Sheet** | **Blue-Sheet** | **Total** | **User_ Accuracy** | **Kappa** |
| Vegetation | 311 | 43 | 21 | 6 | 0 | 381 | 0.816 | |
| Soil | 3 | 468 | 31 | 3 | 0 | 505 | 0.927 | |
| Weakened vegetation | 0 | 6 | 41 | 0 | 0 | 47 | 0.872 | |
| Multisheet and Metal | 2 | 5 | 1 | 21 | 1 | 30 | 0.700 | |
| Bluesheet | 1 | 2 | 0 | 1 | 33 | 37 | 0.892 | |
| Total | 317 | 524 | 94 | 31 | 34 | 1000 | 0.000 | |
| Producer_accuracy | 0.981 | 0.893 | 0.436 | 0.677 | 0.971 | 0.000 | 0.874 | |
| Kappa | | | | | | | | 0.793 |

## 4. Discussion

### 4.1. Difference on the Performances of Mapping Products by the Three Methods

As shown in Figures 8a and 9, the NDVI threshold method could not detect the blue-sheet because it had the same NDVI value range as that of soil (−0.2 to 0.0). In addition, only a part of the multi-sheet in the field was successfully classified, whereas the remaining area was classified as soil. This was because the multi-sheet had been installed in the field for more than two months by the time the aerial surveys were conducted, and the surface was covered by some soil or dust, which resulted in an NDVI close to that of soil.

This result can also be observed in Figure 8. In the area surrounded by the red dotted line, soil was mistakenly determined as multi-sheet or metal. This was because the soil in that area had been stepped on by the surveyors, leaving behind footprints. The water content of the soil compacted by human weight was higher than that of the normal topsoil in the field, which decreased the NDVI value of the compacted area to a level lower than that of the threshold for multi-sheet and metal. This fact indicated that the NDVI value of various soils depends on the soil water content, and misclassification is possible when distinguishing soil from plastic or metal materials simply according to the NDVI threshold. Furthermore, the NDVI threshold method determined the vegetation edges to be weakened vegetation, although these parts were actually green leaves with good health, which can also be observed in Figure 8. This was because the ground resolution of

the multispectral image was more than 6 cm, and the pixels at the edge of the vegetation had average NDVI values of both the vegetation and soil, which made them appear like weakened vegetation. This suggested that despite the UAV multispectral image having a better ground resolution than the traditional aerial photos, there were still error values at the edge of the plant community.

In contrast, as shown in Figure 7b, the RGB image-based machine learning method had a better performance than the former method. This method clearly detected the areas with blue-sheet, multi-sheet, and metal, and classified the compacted soil in the correct class despite the difference in soil water content. However, it still showed a disadvantage in detecting the weakened vegetation, because these areas had similar RGB values to those of soil, and the only standard for classification of this method was the RGB value.

In contrast, as shown by the red dotted line in Figure 7c, the OBIA method successfully detected the area of weakened vegetation which was misclassified by the former method. However, it still showed one disadvantage, which is the misclassification of some weakened vegetation as the normal vegetation class. Figure 6 also reflects this trend of the OBIA method. This was because even though the plants belonged to the weakened vegetation class, the NDVI value might still be similar to that of the healthy vegetation when plant density is extremely high.

### 4.2. Mechanism of the Difference on the Accuracies by the Three Methods

The NDVI Threshold Method achieved high accuracy in classifying vegetation and soil. The user's and producer's accuracies were above 0.80 for both classes. This suggested that the NDVI threshold was appropriate for identifying vegetation and soil. In contrast, the user's (0.200) and producer's (0.309) accuracies of the weakened vegetation class were both low. Furthermore, the user's accuracy was lower than the producer's accuracy, meaning that this method had a tendency to falsely recognize other objects as weakened vegetation. Similarly, both the user's (0.462) and producer's (0.194) accuracies were low for the multi-sheet and metal class, and the user's accuracy was higher than the producer's accuracy. This means that the NDVI threshold method had a tendency to ignore the objects that should have been classified as multi-sheet or metal. Finally, both the user's and producer's accuracies were 0.000 for the blue-sheet class, meaning that the NDVI threshold method did not have the ability to identify plastic material with a rough surface. Considering practicality, the NDVI threshold method demands the least amount of input data, consisting only of the NDVI raster of the field, while it was also the least time-consuming method and could provide an acceptable accuracy in determining the vegetation, soil, plastic with smooth surfaces and the metal material, which made it a practical tool for land cover classification when moderate accuracy was required. As a direction for future research, a more precise threshold value for the classification can lead to a more accurate classification result. In this study and many previous studies, the threshold values were accurate to one decimal place and the threshold value was often defined based on experience. Putra et al. (2015) detected the areas of cloud/water/snow, rocks/bare land, grassland/shrubs and tropical forests/mangrove forest using the Landsat-derived NDVI thresholds of <0, 0–0.1, 0.2–0.3, and 0.4–0.8 [16]. Gross (2005) classified the barren areas of rock/sand/snow, shrub/grassland, and rainforest with the NDVI thresholds of <=0.1, 0.2–0.3, and 0.6–0.8. The NDVI threshold of the non-vegetation was slightly lower than the previous studies, while the NDVI thresholds of vegetation classes were similar. The reason for this difference was that the definition of the non-vegetation areas, such as the bare land, was different between satellite imagery and UAV imagery. The ground resolution of the Landsat imagery was 30 m, which means the pixels that were defined as bare land also contained vegetation; while the ground resolution of the UAV imagery used in this study was less than 0.02 m, which means the pixels that were classified as bare land in this study were pure soil.

For the RGB Machine Learning Method, both the user's (0.904) and producer's (0.946) accuracies of the vegetation class were higher than 0.900, and both the user's (0.877)

and producer's (0.927) accuracies were higher than 0.800, suggesting that this method demonstrated extremely good performance in identifying vegetation and soil in this study area. The user's accuracy (0.720) of the weakened vegetation class was much higher than the producer's accuracy (0.383), meaning that the RGB image-based machine learning method had a tendency to ignore the weakened vegetation. Similarly to the observation and discussion presented in Section 3.2, this result also demonstrated that this method had a disadvantage: mistakenly classifying the weakened vegetation as soil. In contrast, both the user's (0.800) and producer's (0.774) accuracies of the multi-sheet and metal class were very high. Furthermore, the user's (0.971) and producer's (0.971) accuracies of the blue-sheet class were extremely high, meaning that this method could detect plastic and metal materials with very high accuracy. Nowadays, despite the development of UAV remote sensing which provides very high resolution RGB imagery, the classification studies using UAV RGB imagery have still been more focused on land-use classification, instead of ground object classification. Hassan et al. (2011) generated land use/land cover maps including the classes of trees/vegetation, water, soil, urban, and unprocessed area with UAV-obtained RGB images using the supervised classification algorithm (maximum likelihood) and achieved a 90% overall classification accuracy [20]. Shin et al. (2019) conducted classification of forest burn severity with UAV-obtained multispectral imagery using the maximum likelihood and threshold methods and achieved overall accuracies of 89% and 71%, respectively [22]. These results showed that the pixel-based machine learning method could achieve very high accuracy in land cover classification. It is necessary to extend the utility of this methodology from the land-use level to the ground object level. Hamylton et al. (2020) compared the classification results at object level with high resolution UAV GCB images using the pixel classification, visual interpretation, and machine learning approaches, and the machine learning method showed the highest overall accuracy of 85% [21]. Based on the previous studies, the current study has added weight to the fact that the RGB imagery machine learning method is one of the optimal classification methods for ground object classification in rural areas.

For the OBIA Method, similar to the former method, both the vegetation and soil classes had very high user's (0.816 and 0.927, respectively) and producer's (0. 981 and 0.893, respectively) accuracies. The user's accuracy (0.872) was higher than the producer's accuracy (0.436) of the weakened vegetation class. This indicated that although the problem of ignoring the weakened vegetation also existed in the OBIA method, this method achieved a high accuracy in detecting the weakened vegetation, with the best performance among all the three methods. However, regarding the multi-sheet and metal and blue-sheet classes, the user's (0.700 and 0.892, respectively) and producer's (0.677 and 0.971, respectively) accuracies were lower than or equal to those of the former method, suggesting that although the OBIA could detect the plastic and metal materials with satisfactory accuracy, its performance was slightly inferior to that of the RGB image-based machine learning method. The previous studies of this method share the same characteristic with the former method, which is the limitation of application. Natesan et al. (2018) performed land use classification using UAV-obtained multispectral images, and achieved overall accuracies of 78% and 50% for water bodies and mixed-color classification classes, respectively [26]. Ahmed et al. (2017) compared different UAV camera data and platform performance for classifying forest, shrub, and herbaceous layers; bare soil; and built-up areas using the OBIA method and achieved overall accuracies of 90% and 80% with the multispectral camera and RGB sensor, respectively [27]. Sarronet et al. (2018) proposed a method to map individual mango tree production using geographic object-based image analysis (GEOBIA) and obtained an RMSE% accuracy ranging from 20% to 29% [28]. Brovkina et al. (2019) performed forest stand classification with UAV-based NDVI and point dense clouds using the OBIA method and achieved a Kappa coefficient accuracy of 0.74 [29]. Comparison of classification performance between UAV and satellite multispectral image aerial data using the OBIA method by Yang et al. (2019) yielded Kappa coefficients of 0.713 and 0.538, respectively [30]. Ventura et al. (2018) performed mapping and classification of

marine habitats with UAV-obtained RGB images using the OBIA method and achieved an overall accuracy of >80% in different study sites [25]. All these studies focused on land-use classification or vegetation detection. This study has utilized the current methodology for more, smaller research subjects.

### 4.3. Application Values of This Research

The current study has provided a new point of view for the application of the existing classification methodology based on UAV high resolution imagery, which is the classification of various ground objects instead of land-use classification. As mentioned in 4.2, although land-use classification is crucial for many major fields, it does not fully utilize the advantages of UAV remote sensing, which include very high spatial and temporal resolution. The application of the classification methodology should progress with the development of the platform. It is not only increasing the classification accuracy that should be focused on, but also the diversity of subjects. With the suggestions of this study, future research could be enlightened to discuss the ability of UAV remote sensing on detecting more ground objects, including the footprints or individuals of livestock, vermin and wild animals, which contributes greatly not only to the field of agronomy, but also to agriculture, rural development and national park management.

The results of the current study suggested that within the fields of agriculture and natural resources, where vegetation and soil are the main objects of classification, it is recommended to use the NDVI Threshold Method. It demands only multispectral imagery data and has less of a requirement for the image analysis software and technique. In the fields of rural or city environment management, where not only vegetation and soil but also plastic and metal materials are the main objects ofclassification, it is recommended to use the RGB Imagery Machine Learning Method. It demands only RGB imagery data, and could achieve high accuracy with precise imagery analysis with the geographic information system software. Finally, when both the RGB and the multispectral imagery data are available, it is recommended to use the OBIA Method, which could achieve high accuracy in identifying different objects with similar visual appearances.

Based on these result, users of ground object classification, such as societal stakeholders or researchers, from different fields of agricultural and land resources, fishery resources, forests, agroforestry, and ecosystem services would have a clearer standard to decide the optimal classification methods to suit their requirements. Therefore, this study is thought to be helpful for sustainable community development based on the transdisciplinary integration of natural resource management systems. It is also important for communities to work together in the analysis of common UAV imagery to create transdisciplinary collaboration.

Integrated resource management involves multidisciplinary areas, such as forests, agricultural lands, residential areas, lakes, marshes, and oceans, that cover agricultural, land, water, forest, fisheries and tourism resources, agro-ecology, protected areas, ecosystem-based approaches, human well-being, and integrated approaches for the synergistic management of different resources. Although there has been much research dealing with resource management methods focusing on individual resources, there are not many studies discussing evaluation methods that incorporate these resources representing a transdisciplinary approach in a cross-sectional manner. In addition to vegetation such as crops and trees, there are various objects such as plastics and metal wastes on the ground that are subject to resource management. Therefore, the methods discussed in this study for ground object identification are expected to be an important indicator for societal stakeholders and researchers in employing UAV identification methods for integrated resource management.

### 4.4. Limitations and Prospects

The limitation of this study is that the categories of ground objects used in this study were not comprehensive enough to cover every kind of object which can appear in agricultural or rural areas. In discussing the spectral characteristics of other objects, optimal classification parameters such as the NDVI threshold value and the segmentation parameter

for the OBIA method for more objects can be determined, which eventually lead to higher classification accuracy. Furthermore, in this study, the number of training samples for the OBIA method was limited in order to be consistent on the pixel level with the RGB imagery machine learning method for a better comparison. Higher classification accuracy can be achieved by using more training samples for both the RGB method and the OBIA method. Furthermore, the classification classes were limited to natural objects, such as grass and vegetation, which are not enough for practical use in rural areas with artificial objects.

As one of the newest platforms of remote sensing, UAV's advantages are yet to be fully exploited. While the UAV imagery can never take the place of satellite imagery because of the spatial coverage, it provides the highest spatial/temporal resolution and has the highest mobility, which allow the identification of various small objects in local areas. This study has discussed the performance of the UAV imagery in detecting vegetation, weakened vegetation, soil, plastic and metal material, which could be used for identifying crop areas, crop status, waste materials and agricultural tools in a large local area. There are more kinds of ground objects for which automated classification methods have not been discussed yet, such as the footprints of or individual living animals, to contribute to efficient livestock management, injurious animal control and wild animal monitoring in rural areas or national parks.

## 5. Conclusions

This study was conducted to see the applicability and accuracy of the NDVI threshold, RGB Image-based machine learning method and OBIA method using UAV for ground object identification. For this, vegetation, soil, weakened vegetation, blue sheet, multi-sheet and metal were classified and the accuracy of each method was determined.

According to the results of the study, the three classification methods discussed in the present study are based on different technical considerations and exhibited both advantages and disadvantages from certain perspectives.

1.  The RGB image-based machine learning method had the best performance in classifying all types of ground objects in the study area, whereas the OBIA method had a slightly lower overall accuracy and the NDVI threshold method had the lowest accuracy among the three methods.
2.  The NDVI threshold method demands the least amount of input data, only requiring the NDVI raster of the field, while it was also the least time-consuming method and could provide acceptable accuracy in determining the vegetation and the metal material.
3.  The RGB image-based machine learning method had better performance at detecting plastic and metal materials, which had bright RGB colors.
4.  The OBIA method had better performance at separating objects with similar RGB characteristics but different multispectral reflectance characteristics, such as for soil and weakened vegetation.
5.  By verifying and comparing the performance of the existing classification methods on detecting various objects, this study unraveled the mechanism of the difference of the classification accuracies by the three methods, and made recommendations for UAV users from different fields of the optimal method, which is thought to be a contribution to transdisciplinary integration.

**Author Contributions:** K.Z. designed and conducted the surveys, analyzed the data, and wrote the manuscript. S.M. participated at discussing the manuscript construction, and revised the manuscript. H.O. conceptualized this research and contributed greatly to manuscript revision. A.S. and S.S. gave advice about the data analysis methods and the manuscript construction, and revised the manuscript. K.H. supervised the manuscript and made significant comments to improve it. T.H. contributed greatly to the manuscript and participated in its revision. L.F. provided background knowledge and revised the manuscript. All authors have read and agreed to the published version of the manuscript.

**Funding:** This study was funded by the "Establishment of a Sustainable Community Development Model based on Integrated Natural Resource Management System in Lake Malawi National Park

(Int NRMS) Project" under the Science and Technology Research Partnership for Sustainable Development (SATREPS) program supported by the Japan Science and Technology Agency (JST) and Japan International Cooperation Agency (JICA). This work was also supported by JSPS KAKENHI Grant Number JP20K06351 and research programs of the Tokyo NODAI Research Institute, Tokyo University of Agriculture, and the Institute of Materials and Systems for Sustainability (ICMaSS), Nagoya University.

**Institutional Review Board Statement:** Not applicable.

**Informed Consent Statement:** Not applicable.

**Data Availability Statement:** The data presented in this study are available upon request from the corresponding author. Interested researchers can contact the corresponding author directly through e-mail.

**Acknowledgments:** This paper benefited greatly from discussions with Int NRMS project members, especially those of the Agriculture Resource Management Group of the project. Atsuko Fukushima (Ehime University) provided administrative support throughout the research processes. This research would not have been completed without their help. The authors are grateful to Osamu Tsuji and Masahiro Akimoto (Obihiro University of Agriculture and Veterinary Medicine) for providing the field for this study. The authors would like to thank those individuals who gave them the technical idea and analysis guidance which became the motivation for this study. This research would not have been completed without the help of these individuals.

**Conflicts of Interest:** The authors declare no conflict of interest.

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
