# Peer review of "Assessment of Three Automated Identification Methods for Ground Object Based on UAV Imagery"

_sustainability, doi:10.3390/su142114603_

Round 1
Reviewer 1 Report
Article describes automated identification of ground objects as folia, and also soil structure, plant vegetation performance, what is very practical in farming practices, but esspecially before planing of field activity. It is important to know not only the plant vegetation level, but also plant health itself.
Please add section 5. Conclusions with subpoints describing the most important findings of the study. Such summary will give the article more readible in the aspect of research evaluation phenomena.
Please add some recent references concerning automated identification methods monitoring ground farming objects.
Reviewer 2 Report
The topic selection is meaningful. However, the documents I have seen so far, the results, discussions and conclusions are so mixed in form that I cannot make a clear judgment. But I would like to give some suggestions as below first and review later when I see the corrected manuscript.
Several problems need to be considered:
1. In the introduction, 1.1 and 1.2 can be combined, but the necessity of writing them separately is not fully reflected at present.
2. It is suggested to discuss the limitations and future research directions.
3. The paper should explain the underlying remote sensing mechanism of the different accuracy of the three methods.
4. I suggest that the paper discuss the application potential of the research.
Reviewer 3 Report
Dear Authors, I don't see any significant improvement in the manuscript, I still think it is nothing more that a technical exercise and that it can not be published as a scientific article. Moreover the revised version is very confusing (discussion with not well formatted table, then conclusion then another time discussion in track change mode). Finally, I appreciate the efforts of authors in justifying the application of UAVs in such kind of studies but I still believe they are not suitable for this kind of application and therefore I can't see the benefits of this manuscript also from the operational point of view...
Reviewer 4 Report
This research discussed the topic of the ground object identification by using UAV imagery, but author did not present some new method and the work is not novel. And this research has little reference value for readers in the field.
Some specific comments:
1. The study is more like an experimental report. Although several methods are compared, they are all common methods and not novel.
2. The types of ground objects in the research region are simple. But some of the objects are very special, such as multi-sheet and metal and plastic blue sheet. But these objects can not be identified without the help of the spectral feature and texture feature in a complex condition of the real world.
3. The reflection characteristics of ground objects need to be analyzed, especially for some special object.
4. The principle or rule, which used to distinguish the healthy and dead/weaken-end vegetation, just based on NDVI? Some of the healthy vegetation with low vegetation cover also has very low NDVI value. Some of the spectral analyze are needed.
5. The time of the UVA data collection?
6. Line175-177 is the aim of the research, I think it should not present in this section.
Round 2
Reviewer 2 Report
Thank you for the revision and please check the attachment file.

Author Response
Dear reviewer,
Our sincere gratitude for your comments and advice for this paper.
Please check the attachment.
Thank you.
Authors

Reviewer 3 Report
Dear Authors, with the changes made the paper has been improved. I leave the final decision to the editor
Author Response
Dear reviewer,
Thank you again for confirming the manuscript and report.
Our sincere gratitude for your time.
Authors
Reviewer 4 Report
To Author,
I found some of the suggestion has been revised and I have no other comments to the authors.
Author Response
Dear reviewer,
Thank you again for confirming the manuscript and report.
Our sincere gratitude for your time and efforts in improving the paper.
Authors
Round 3
Reviewer 2 Report
Thanks to the author for the revision. I would like to give one more comment: Limitations and prospects can be considered as a seperate section (4.4) in discussion.
Author Response
Dear reviewer,
Thank you very much for the proper advice about section 4.4.
The change has been made as you advised, and it did make the Discussion clearer and more ordered.
The authors thank you very much for being so conscientious about the structure of this paper. We have learnt a lot during the revision process, and will use the experience in our future work.
Our gratitude again,
Authors